# Emergent Strategy in Higher Education: Postmodern Digital and the Future?

Mohamed Ashmel Mohamed Hashim [1] , Issam Tlemsani [2] , Robin Matthews [3], Rachel Mason-Jones [1] and Vera Ndrecaj [1,*]

1  Cardiff School of Management, Cardiff Metropolitan University, Western Avenue, Cardiff CF5 2YB, UK
2  The Centre for International Business, TCIB, Omega House, New Malden, London KT3 6DR, UK
3  London School of Commerce, Chaucer House, White Hart Yard, London SE1 1NX, UK
*  Correspondence: vndrecaj@cardiffmet.ac.uk

**Abstract:** Mintzberg's version of emergent strategy is based on the idea that strategies are contingent on circumstances that change from time to time often very rapidly and therefore papers focused on strategy and detailed planning are limited in their practical application. The word strategy as far as Mintzberg is concerned is anathema, therefore, introducing a concept that has a misconception embedded in it. This paper claims that education for sustainable development and higher education institutions' survival depends on adopting postmodern thinking, in other words, digital transformation. This conceptual paper proposes a blueprint of a process for developing a series of agile potentially short-term conceptual solutions thereby embracing the expectation that the rate of change in societies is accelerating. This paper scrutinizes (a) the applicability of emergent strategy/strategic approach to higher education institutions, (b) how postmodernism influences higher education institutions to become digital hubs of commoditization of knowledge and (c) how the integrated capabilities of digital transformation build sustainability in education delivery. Structural Equation Methodology is proposed to examine the impact of postmodernism on the sustainable delivery of education in higher education institutions, and the need to foster relevant emergent strategies is also justified. The paper also develops new research propositions and managerial implications for driving optimistic digital education. Ultimately, it offers a framework for spear-leading effective and leading post-modernistic digital transformation. Emerging education technology, sustainable digital transformation and advanced use of robotic-human cognitive collaboration are experiencing a significant transformation. Universities play a vital role in enhancing engagement within higher education. One of the managerial implications of the results and discussion is the need for higher education institutions to provide taught leadership and planning in emergent strategy formulation and implementation. The findings confirm the significant importance of linking the Structural Equation Method and the postmodern strategic context in which we argue that higher education institutions require emerging rethinking.

**Keywords:** postmodernism; digital transformation; DAO and emerging university strategy; sustainability

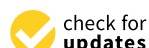



## 1. Introduction

Higher education institutions in the 21st century face a competitive landscape that has changed entirely in the last 40 years or so. Indeed, the adjective 'competitive' has shifted from the ancient rivalry based on academic reputation to peaks to a correlated landscape in multiple versions.

Higher education institutions have become distributed autonomous organizations (DAOs), partly by design but mainly because of the fragmentation of the competitive landscape and their own almost involuntary internal adaptive digital processes. The existing narrative no longer captures their fragmented competitive landscape (Cabrera et al.

2022; Garrod 2016; Wang et al. 2019; Bellavitis et al. 2022). Fragmentation has occurred because of a variety of contributing factors such as the emergence of Massive Open Online Courses (MOOCs), the trend towards blended learning, reinforced by COVID-19, information exchange, commoditization of knowledge, and strategic decision making, which is distributed among internal and external influences: the ingredients of a postmodern situation (Player et al. 2020; Raviolo 2013; Manning 2012). Higher education institutions tend to invest in digital transformation strategies in order to be prepared for the pressing challenges of globalized education (Mohamed Hashim et al. 2021). Radical postmodern changes in global education have enabled higher education institutions to develop sustainable digital transformation strategies to stay competitive. What does it mean to stay competitive in global education? Staying competitive in the global education landscape demands long-term strategies to coup-up with the postmodern challenges. How can higher education institutions use sustainable digital transformation strategies to attain sustainability in education delivery? There is a trade-off between cost implications and various facets of achieving sustainability (Grenčíková et al. 2021). We set ourselves principal tasks in the paper. First, to elucidate the current situation higher education institutions are in and second, to explore and identify a scheme or framework for adaptation processes.

Postmodernism is a phenomenon that has developed a stronghold in higher education. The depth of critical exploration is found in the writing of established scholars (Lyotard 1984; Clark 2006; Richardson and Jencks 1989; Kahraman 2015; Lyotard 1984). This phenomenon has considerably influenced the sustainable delivery of global education and its landscape and is closely associated with the unique phenomenon of sustainable digital transformation. Digital transformation in the global higher education industry determines the future roadmap to a sustainable education management strategy. Thus, there is a need for higher education to develop emergent education strategies integrated with the forces of postmodernism (Vica Olariu et al. 2020; El Kamel and Rigaux-Bricmont 2011; Kahraman 2015). Table 1 presents a detailed examination of literature and influence voices exploring postmodern forces of education.

**Table 1.** Key features of postmodern society and digital transformation. Source: Based on dwellcc.org (2020), Richardson and Jencks (1989), Kahraman (2015), Lyotard (1984).

| Key Features | Features Education of Modern Society | Changes in Features of Postmodern Society | Role of Digital Transformation | The Impact on Education Sustainable Delivery |
|---|---|---|---|---|
| The form of Knowledge | Essentially is controlled by an authoritative mechanism and unbiased knowledge. | Biased knowledge and the higher education institutions' educators are the architects of biased/ new knowledge. | Academic Program Management/Review/Monitor and control. | Regulate the academic delivery. |
| Spending pattern. | Spending is controlled and approved by the state. | Independent of spending priorities, but justification is required. | Virtual planning, communication, and coordination of the academic programmes. | Track the delivery progress/gain visibility of the key global changes in education. |
| Commoditization of education. | Education is fixed—time, place, and cost. | Types of choices for selecting higher education institutions—virtual, online, distributed and distance learning. | Offers students various options to follow and complete the course through information technology education tools. | Virtual, online, distributed and distance learning. |

**Table 1.** *Cont.*

| Key Features | Features Education of Modern Society | Changes in Features of Postmodern Society | Role of Digital Transformation | The Impact on Education Sustainable Delivery |
|---|---|---|---|---|
| Change. | Lecturer/teacher lead. | Independent learning, teachers are there to guide and set up the challenges. | Cases, sums, and challenges are posted to the students online | Offer the opportunity for blended learning/ new pedagogy |
| Culture and Values. | Unique, students are expected to learn the culture; it also can be viewed as a barrier. Attempt to be value neutral. | Unified learning society and education build diverse personal values. | Digitalization promotes unity among students. Thus, unity is based on the dominant digital culture in education. | Digital delivery creates/develops an equal culture. |
| Student nature and the curriculum. | The objective is to meet the national curriculum. | Complex, it needs to meet the needs of globalization of education. Responding to meet global, social, economic, and political pressure. Move from the national to the global context. | It has become the common platform for delivering global-ized/commoditization of education. | Enable higher education institutions to deliver the courses according to global delivery standards such as AQA and AACSB. |

This conceptual paper aims to develop a conceptual model for implementing post-modernistic digital transformation in higher education. The model advocates how digital transformation can act as an enabling force to develop competitive advantages for higher education institutions in the context of postmodern education (Morze and Strutynska 2021). Building competitive advantage is a relative, evolving, and important concept in strategy formulation. In recent years, specifically in the education industry, the notion of building competitive advantage has been challenged by global phenomena such as digital transformation, globalization, information exchange, digitization, and social media in most global industries. These phenomena have collectively made the process of building a competitive advantage in a rapidly changing, short-term landscape (Abad-Segura et al. 2020; Akhmetshin et al. 2020).

The emergent strategy/approach has become increasingly important in global educa-tion because of its ability to deal with inevitable changes such as the impact of postmod-ernism, digital transformation, and sustainability of delivery (Foss et al. 2021; Mirabeau and Maguire 2013; Davies and Walters 2004; Kahraman 2015). The successful implementation of an emergent strategy is broadly recognized in global industries. Seasoned education strategists and entrepreneurs can teach us a lot about the need for emergent strategy and how to best approach it. The global education industry is evolving, it is typified by key features such as innovation, transformation, and agility. Thus, higher education institu-tions face various challenges in establishing a model to build an emergent strategy while systematically integrating the influence of postmodernism and digital transformation. On this notion, we propose a unique emergent approach for education strategy- an emergent strategy for education using design thinking. Thus, this paper critically reviews (a) the need for emergent strategy, (b) the integration of postmodernism-digital transformation and (c) the sustainability of the digital delivery of education. Higher education strategy endures a prime responsibility for establishing competitiveness, economic performance, and shaping graduates' futures. Education strategists embark on formulating emergent strategies for higher education institutions to cope with the changing global education landscape/market conditions (Mintzberg and Waters 1985). Thus, there is a significant need to establish a practical approach to emergent strategy (Fixson and Rao 2014).

Mintzberg (1978), the chief architect of the emergent strategy, argues that the intended strategy does not necessarily come into a realisation. Thus, it becomes an unrealised strategy. Hence, there is a need to understand the realised strategy using an empirical approach while highlighting outside the planning activities/process. In this context, the realised strategy results in patterns of activities the management does not anticipate (Vica Olariu et al. 2020; Hernández-Betancur et al. 2017), this resulting response approach is titled an emergent strategy. Figure 1 illustrates the emergent strategy process. There are four factors that define the current situation that higher education institutions find themselves having to respond to, these being (a) the eruption of accelerating technological change, (b) expanding range of product attributes, (c) internationalisation and conflicting government policies and (d) that define the current state of higher education institutions. This paper explores is the role emergence of digital technologies and the information revolution on the resulting fragmented postmodern landscape.

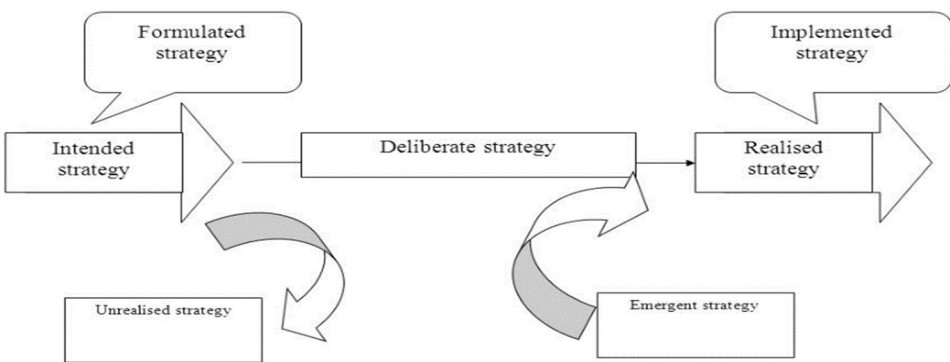

**Figure 1.** Pattern in strategy formation. Source: Based on H. Mintzberg: "Management Science May 198; 24, P.945".

Higher education institutions can explore new business opportunities by going beyond the traditional approach to strategies, tools, and changing market conditions. Specifically, the influence of postmodernism, globalization, digital transformation, and information exchange have rapidly changed global education. Thus, the significant need for an emergent strategy/strategic approach is realised. Using design thinking, the simplistic approach to emergent strategy adopts an incremental mechanism, which is demonstrated below in Figure 2 the incremental-act model.

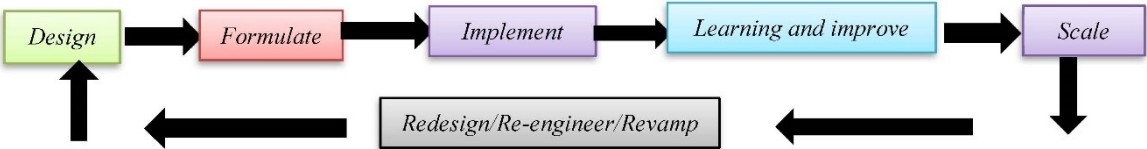

**Figure 2.** Fundamental model of emergent approach to education strategy. Source: Based on Fixson and Rao (2014).

The higher education institutions that adopt an emergent strategy approach do not only/constrain themselves to critically analysing historical data to predict the future/via conventional forecasting. Instead, they capitalize on market opportunities based on predicated changes by taking calculated risks using reliable and scalable steps using an experimental approach, examining and evaluating the outcome of each step.

Adopting a robust lesson learnt approach allows a structured approach to formulating measured actions. Meaning each step forward reveals the previously covered challenges and baseline for the next step- thereby ensuring the notion underlying the emergent strategy. We argue that one of the key problems of the prescriptive/analytical approach to business strategy is not the mechanism but rather the unreliability of tools predicting uncertainties

specifically in terms of application and understanding potentially good and bad scenarios. For example, statistical analysis of return on investment- discounted cash flow provides stakeholders with false interpretation certainty about the naturally uncertain condition. On this notion, we claim the emergent approach becomes a necessity for education strategy amid the agile changes of postmodernism and digital transformation.

Postmodernists argue that increasingly societies are characterised by consumerism and choices. The influence of postmodernism challenges global education to explore beyond the conventional operations of higher education institutions and conventional education delivery, which favours liberal education. We claim that we still live in a postmodern society/postmodern age which typified five major characteristics but not limited to (a) diversity of individuals, (b) better fluidity in identity and appearance, (c) emergence of cross-culture, (d) globalization of education, (f) commoditization of knowledge and (f) media-saturated life (El Kamel and Rigaux-Bricmont 2011; Emerick 2007; Nielsen 2006; Hassard 2003).

What do these changes mean to global education? How does it impact the delivery of university education? Why does digital transformation become inevitable in the delivery of education? What impact would it have on the sustainability of education? These are critical questions in the age of information exchange.

The key characteristics of postmodern education society are shown in Figure 3, although its significance and impact are unclear, specifically in global education. Relatively it is under investigation, and a paucity of knowledge is evident in the literature. We favour the conventional of wisdom postmodernism to enrich contemporary education society, verify the compatibility of its key features and validate the need to establish rigorous organizational research. Global education is closely associated with the information revolution powered by digitalization and technologies. Almost by definition, digitalization leads to parallel computing, which leads to the emergence of DAO's fragmentation and hence postmodernism.

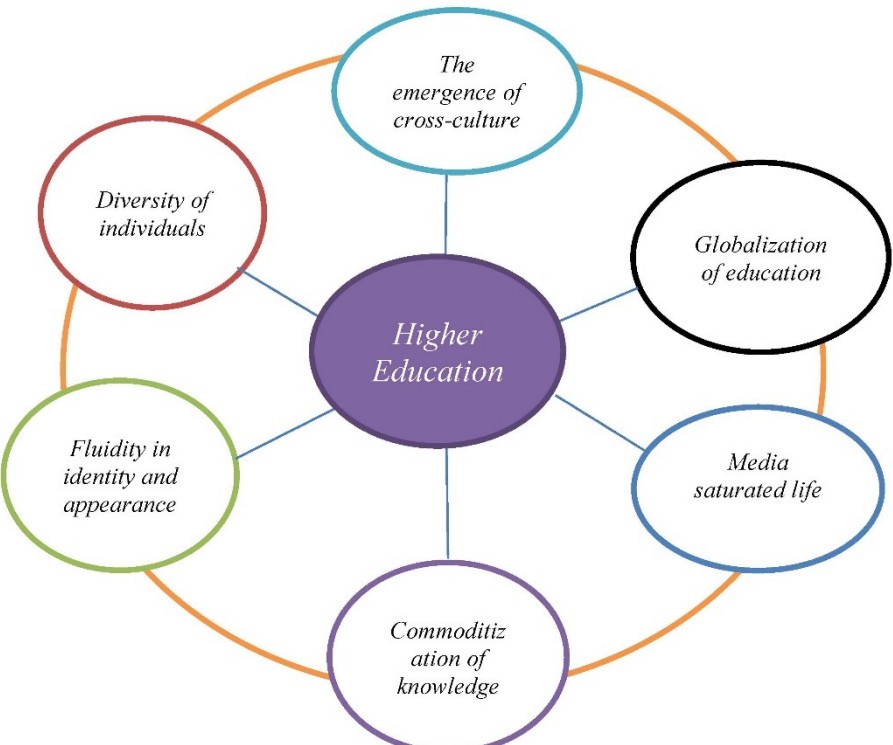

**Figure 3.** The key characteristic of postmodern education society. Source: Based on Lyotard (1984), Richardson and Jencks (1989), Kahraman (2015), and Authors' proposal (2022).

Figure 4 indicates that the emergent strategic approach to higher education becomes a necessity because of the significant influence of real-world phenomena, namely (a) the influence of postmodernism and (b) digital transformation shaping the sustainable delivery of education. Despite the collective impact of postmodernism and digital transformation on education and its wide acknowledgement, there is limited of knowledge regarding how it puts pressure on higher education institutions to deliver education sustainably by amalgamating delivery models. This body of literature states that digital transformation act as a proxy between the way university integrates the impact of postmodernism into their digital transformation strategy, in turn, its impact on the delivery of sustainable education/its delivery (Usher and Edwards 1994; Klimski 2018; Vica Olariu et al. 2020; El Kamel and Rigaux-Bricmont 2011; Kahraman 2015). To fill this gap in this literature, this research explores (a) how postmodernism impact sustainable digital transformation and (b) how digital transformation influences sustainable education of higher education institutions.

**Figure 4.** The fundamental model of postmodernism on sustainable digital transformation. Source: Based on the author's proposal (2022).

How should higher education institutions shape and reshape their education delivery amid the pressing changes of postmodernism and digital transformation to foster sustainable education delivery while coping with the growing demand of the age of globalization? Digital transformation, agility in education delivery, the resilience of higher education institutions, blended education, and relevant affiliation are continued to be the vital elements of the postmodernism of higher education and the higher education digital future.

At this juncture, it is critical to examine the key changes in education from a modern society to a postmodern society, particularly how it evolves. What does it mean to the future of education, and how higher education institutions can make sense of it? What is the role of digital association? How is it intrinsically interrelated with postmodern education society? In order to answer these questions, first, it is essential to identify and determine the key changes caused by postmodernism and how it is closely associated with the process of digital transformation as far as university education is concerned (Usher and Edwards 1994; Klimski 2018). Table 1 presents an overview of the key features identified within the literature.

We argue, comparatively, that there is a paucity of knowledge in terms of how postmodernism changes to sustainable digital transformation-based education globally. Further, how this change leads to sustainable education delivery at the university level has had limited exploration in the literature at the university level. It is therefore, posited that the relevance of Postmodernism on sustainable digital transformation and how it impacts the need for sustainable education requires significant investigation both globally and locally (Kia 1988).

The higher education institutions need to examine how the impact of most modernism influenced emerging changes in the delivery of sustainable education. The demand for the globalized sustainable delivery of education has enabled higher education institutions worldwide to demonstrate critical features such as (a) flexibility, (b) differentiation, (c) agile, (d) mobility and (e) decentralization while design-developing and delivering portfolios of educational courses (Mohamed Hashim et al. 2021, 2022; Lozano et al. 2015) Meaning, as a community, we are transitioning into a new educational era, and inevitably, the phenomena-impact of postmodernism and sustainable digital transformation- are collectively changing our educational balance. Within this examination we need to explore a couple of key questions, such as: Are we living in a new type of educational landscape? If so, what kind

of landscape is it? In order to explore these questions, we require constructive dialogue to enable effective critical exploration.

This paper attempts to bridge an existing research gap by (a) developing the existing body of knowledge about postmodern education to the next level, (b) critically examining the close associations between postmodern education, digital transformation, and the need for emergent strategy and (c) developed practical conceptual models for assessing the realistic impact of postmodernism on sustainable education delivery in higher education institutions. As stated, this paper aims to offer a relatively implementable model for post-modernistic digital transformation in higher education.

The authors have developed the following key research questions to achieve the aim.

(a) How postmodernism of education impacts the sustainable delivery transformation of higher education institutions.
(b) What forces influence the postmodernism of education and sustainable delivery of education?
(c) How international collaboration integrates with the impact of postmodernism on sustainable education delivery.

## 2. Literature

The literature review drew upon the existing literature exploring (the emergent strategy process based on Quinn (1980), Senge (1990), Argyris (2014), Mintzberg (1987) and Lynch (2018). Whittington et al. (2020) empirically demonstrate how organizations seize incremental, adaptive, flexible, experimental, and learning and development while design-developing, implementing, and re-engineering business strategies.

This conceptual paper claims that developing a strategic emergent approach that interacts with postmodernism and digital transformation capabilities (Figure 5), in turn, enables higher education institutions to gain sustainability in the delivery of education. It needs to be highlighted that in the age of globalization, the critical success factors of university education are increasingly standardised. Thus, the notion of building competitive advantage requires thinking beyond conventional structures and resourcing approaches within the education industry, it requires unique models, processes, design thinking and selective integration capabilities. The subsequent sections critically examine the formation/approach to emergent strategy for education, the importance of integrating the global influence of postmodernism and the utility of digital transformation on university education and how it enables building sustainable delivery models for education. Specifically, the phenomenon- Emergent Strategy in Higher Education: Postmodern Digital and the Future? Require thorough examination/investigation to use knowledge in the educational society and strategic management of education as to how to build, retain and protect the process of building competitive advantages (de S. Oliveira and de Souza 2021; Teece 2020; Halliday 2020; Lamichhane and Wagley 2013).

### 2.1. The Emergent Approach of Education Strategy

The robustness of strategy tends to be thought of as forceful, flexible, interactive and based on learning and development. Therefore, the concept of enabling the emergent strategy is increasingly becoming important in the rapidly changing landscape of global education. However, the application and integration of the emergent strategy approach lack significant theoretical underpinning, specifically in terms of how to interact in the dynamic education environment and leverage it to create superior value. The emergent strategy approach offers education a way to achieve sustainable education delivery. We provide a theoretical foundation and a unique approach by amalgamating postmodernism and digital transformation educational changes. An emergent strategic approach is wide spreading, specifically in the global education industry. Specifically, education stakeholders are comfortable with the notion/logic the emergent strategy underpins. However, it can be shown there is limited understanding of the approach and especially its practical application within managerial levels. One could argue that the formulation of an emergent

strategy requires the identification and determination of uncertainty within the macro and micro-environment as the first step in the current education industry (globalised and post-modernised industry). Thus, the stakeholders who closely interact with strategy formulation and implementation require sensing skills, corporate games knowledge, and simulation practices.

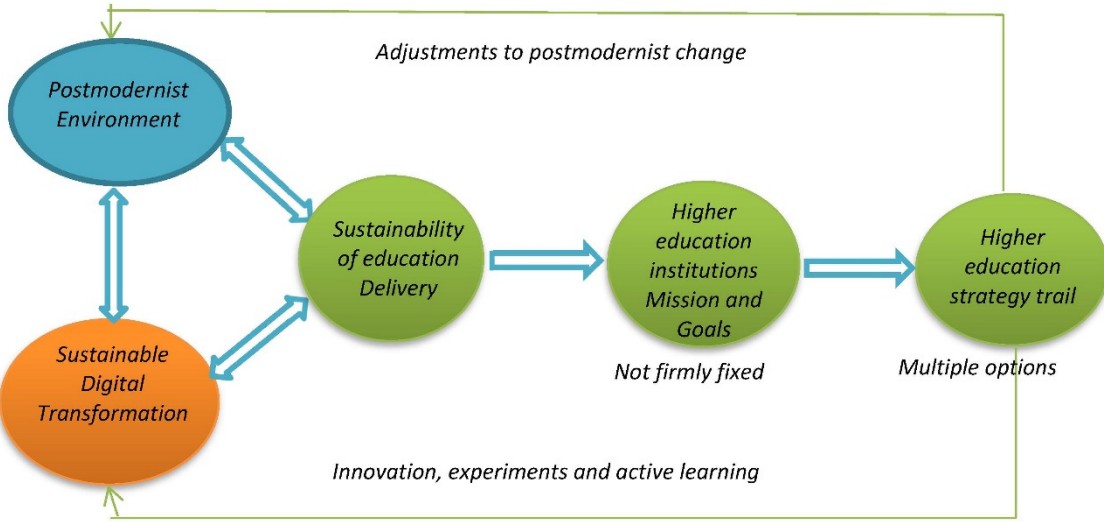

**Figure 5.** The postmodern-digital future. Source: Based on Argyris (2014), Mintzberg (1987), Quinn (1980), Senge (1990), Lynch (2018) and Whittington et al. (2020).

The application of an emergent approach to education requires (a) continuous knowledge acquisition, (b) the use of a reliable method, and (c) an organizational mindset to strive in global education. The emergent strategy implementation empowers higher education institutions to develop a portfolio of educational offerings at various levels through experiential learning. Further, the implementation of an emergent strategy also enables the lecturers, education administrators and students to learn from the unsuccessful experiments (Morze and Strutynska 2021).

*2.2. Postmodernism*

Lyotard (1984) on postmodernism claimed that postmodernism is due to people's wariness and disbelief regarding the metanarratives- therefore no longer believing there is only one truth or mechanistic solution. Postmodern society stops blind belief in the big stories and is more open to an exploration of contextual understanding and adaption. Instead, individuals are more open to develop their unique perspectives on events and indeed understanding the challenges of variation of interpretation to move towards a collective understanding of the wider narrative. This feature of postmodernism has impacted the design and delivery of education.

Today higher education institutions are under increasing pressure to impart one-to-one engagement mechanisms and indeed experience while delivering courses. Irrespective of the generic course delivery, stimulating one-to-one engagement experience is becoming a fundamental need for higher education institutions, digital transformation platforms this experience primarily because of its customisable ability to generate various solutions (Leaning 2014; Pivovarova et al. 2020; Lamichhane and Wagley 2013; Zhu 2009; Done and Knowler 2013).

So, what do postmodernists believe in? We assess that postmodernism is typified by the distrust of educational experts, meaning that students in the modern era believe in more than one truth in selecting and pursuing a course while still understanding some core truths that exist. They validate the authenticity through various measures. In other words, students are potentially more engaged with the concepts of critical discussion. For

example, understanding phenomena such as COVID-19 and climate change crises cannot be distilled down to one universal solution but is recognised to have a variety of complex contributors and requirements in terms of effective response. It is also a noticeable fact that while pursuing education, students are willing to remain as global citizens, and their structural identities, such as nationalities, tribes, classes, and ethnicities, are becoming less important.

Usher et al. (1997) state that education in the postmodern society is explained and shaped by diversity. Meaning education delivery must provide the learners with lots of choices. Robin Usher and Richard Edwards were the pioneers who studied the empirical relationship between postmodernism and education in a global context. Specifically, they stated that postmodernism and its influence are significantly important for university-level education.

Education institutions should offer courses that suit the learners changing needs, which may influence by the changing economic, social and political situations. It should mean that global education reflects the postmodern economy or service economy. This phenomenon has created desirable opportunities for higher education institutions to offer education in the forms of distance learning, virtual learning and blended learning. Thus, the digital transformation of education becomes inevitable. Particularly, it has become a centric feature of postmodernist education life. It is important to highlight the key features of postmodernist education at this juncture.

Genosko (2001) quoted Baudrillard's view on postmodernism in the year 2021 stating that postmodernism prioritises individualism (choices and opinions) relative to socialism. Further, some scholars argue the birth of postmodernism leads to the death of socialism. What does this mean to the higher education institutions' education system? Increasingly, higher education institutions' signs and symbols have increasing importance. It is argued there are no standard key performance indicators to validate if the required values (or indeed achieved) are reflected in the delivery of education. Value t is quite hard to differentiate between reality and hyper-reality.

We argue that higher education institutions are increasingly building international collaboration and gaining accreditation to maximise the educational values communicated by the sign, symbols, and badges. Often collaboration is used to power the accreditation of degrees, and higher education institutions use it as a shield to attract revenue and fill any potential revenue gap. In the global education landscape, the standard of education accreditation impacts the delivery models and hence potential profitability. When higher education institutions gain multiple accreditations, it enables them to tailor the education delivery while building the flexibility to actively engage students in pursuing their expected learning outcomes and career aspirations (Mohamed Hashim et al. 2021).

The other key distinguishing feature of postmodernist education is hyper-social media saturation, which builds to hyper-reality. The images and logos of educational institutions often describe the values/experience delivered to the students. We view that as it is an illusion of reality-simulacrum. Today, what is unreal is perceived as real because of social media reality. Education institutions excessively use simulacra to which students believe with the stimuli until they no longer believe in the reality of education. At this juncture, a key critical question is what the role of social media in global education continues to be unclear but complex.

This is most evident across Facebook, YouTube, Instagram and Twitter. Social media personalities/superstars are increasingly used for branding/attracting students for various education portfolios. Specifically, this is excessively evident in fashion design courses. At a high-level, images of celebrities are presented selectively in social media to paint the illusion of reality. As part of their strategy, most of the university's practices distinguish the reality of students and what they perceive from the social media paints about the delivery of education. The illusion built via social media fails to materialise real meaning to students' education life and experience.

Students choose courses based on industrial demand. Thus, there is a danger that the philosophy of wider education choice may become fragmented/unstable if universities focus too much attention on current trends. This phenomenon has imposed noticeable volatility in identifying and determining the portfolio of courses to be offered. Thus, higher education institutions are under considerable pressure to understand the key changing education trends. To be very specific, there is no fixed formula to determine the portfolio of courses. As factually justified, we live in a society where the influence of postmodernism is immense. It has broken down the education society into individual narratives recognising multiple identities, volatility, complexities, and potential confusion (Lyotard 1984).

The rationality of postmodernism is important to transformative education. In the age of globalization, it is achieved via the digital transformation of education in higher education institutions, as it establishes the notion of theoretical basis, the foundation of digitalization of university education/learning experience, importantly; this is relevant to the digitalization of sustainable education/education delivery-an increasingly popular subject in university education. This paper engages with the future of university education as an interdisciplinary with postmodernism and digital transformation. We argue, what could be viewed as important of postmodernism in gaining academic respectability? (Holsberry 1981; Lyotard 1984).

In the age of postmodernism, the delivery of global education is characterised by but not limited to (a) digitalization of education, (b) education as a service, (c) commoditisation of education, (d) privatisation of education (e) marketization of education (f) virtual learning, (g) independent learning, (h) decentralised learning (i) social media-saturated education, (j) scientific thinking, (k) accreditation is influenced by distributed education, (l) standard of education and (m) quality of education. The collective and serious changes of postmodernism problematize and disrupt the deep-rooted assumptions of university education, specifically teaching, learning, and delivery. This conceptual paper concludes what university education might become due to post-modernistic disruption and turbulence.

### 2.2.1. Digitalization of Education

Higher education is globally undergoing significant changes, which are primarily influenced by the impact of postmodernism and technological advancements. Particularly, the influence of postmodernism has enabled higher education institutions to adjust their educational deliverables using innovative delivery models. The deliverables are tailored according to the knowledge of economic changes; the digital transformation is used as a tool/platform to create value the educational delivery (Bican and Brem 2020; Benavides et al. 2020; Abad-Segura et al. 2020; Bogdandy et al. 2020; Akhmetshin et al. 2020; Iivari et al. 2020).

This approach enables higher education institutions to take a sustainable approach to global education delivery. We argue that the influence of the postmodern digital transformation and how it impacts education sustainability should be viewed as a key change in the socio-economic education system. Further, other factors such as globalization and information exchange are fuelling the key characteristics of global education ((a) digitalization of education, (b) education as service, (c) commoditisation of education, (d) privatisation of education (e) marketization of education (f) virtual learning, (g) independent learning, (h) decentralised learning (i) social media-saturated education, (j) scientific thinking, (k) accreditation is influenced by distributed education, (l) standard of education and (m) quality of education).

Higher education institutions worldwide use digital transformation as an alternative to fill the student enrolment gap. This is a common feature of the digital transformation strategy, relatively influenced by the shock of postmodernism. This unique education phenomenon has enabled higher education institutions to examine the sustainability of their education delivery. However, this realm is relatively still in the embryonic stage, drastically different in scope, and requires rigorous investigation. We view that the empirical insight

of this research leads higher education institutions to build entrepreneurial capabilities, which can be translated into developing competitive advantage in the long run.

2.2.2. Commoditisation of Education and Virtual Learning

Commoditisation of education has provided a strategic option for higher education institutions to equalize work experiences to degree-level standards. As stated, the commoditisation of education, which is one of the key features of postmodernism, has aided higher education institutions in attracting qualified faculty members in relevant and demanded fields of education.

The pressure of commoditisation of education in higher education institutions' planning also potentially fueled resources to be viewed as a commodity to fulfil the delivery commitment of education. This situation diminishes the strategic importance of faculties in the role of delivery. Thus, higher education institutions explore alternative delivery methods and unique delivery models to meet the demands of the commoditisation of education.

In the postmodern environment, we argue that the average life span of education delivery (actual delivery) is shrinking. The power of digitalization also reduces and optimizes the lifecycle times or patterns. Thus, faculties go through specialised training to cope with the need for the digital transformation of education. We argue that education's virtual commoditization has already set a new standard for faculties and students. Utilizing digital transformation capabilities to meet delivery needs goes beyond technical understanding and requires a deeper understanding of pedagogical preparation in the digital environment (Carter et al. 2020; Chambers 2016). The benefits of Virtual learning are widely acknowledged. As indicated by Figure 6 it attempts to enhance collaborative and engaging learning. A standardised virtual environment is typified by three distinct features, namely (a) physicality, (b) interactivity and (c) persistence. The participation and engagement of the students are represented by digital/graphical representation (Bican and Brem 2020).

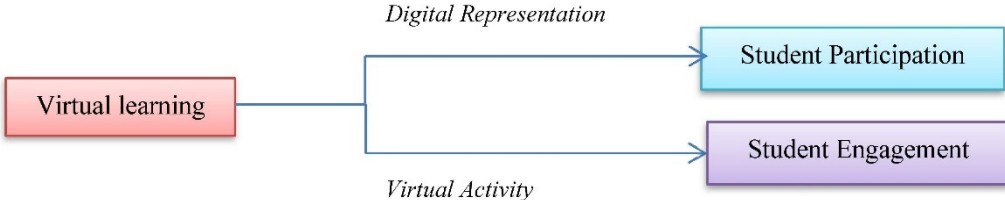

**Figure 6.** Key components of virtual learning. Source: Based on Al-Azzam et al. (2020), Johnson and Blitzer (2020), and Authors proposal (2022).

Virtual learning is a significant component of student learning approach pedagogies and has been fueled by the response to COVID-19. Virtual Learning (VL) enables students to access content/videos/presentations anytime via multiple communication channels. Modern-day virtual learning tools' innovative features and capabilities assist learners in engaging in profound interactions and have close engagement experiences; otherwise, they would have in classical face-to-face learning. Arguably, virtual learning is taking the centric orientation of blended learning. The use of digital space is a desirable feature (Henseruk and Martyniuk 2020).

Inevitably, virtual learning has become a propelling force for higher education institutions, particularly because of COVID-19 (Al-Azzam et al. 2020; Johnson and Blitzer 2020). Thus, the authors argue that higher education institutions must extend the utility of virtual learning to instrumentalize the sustainable delivery of education. Therefore, higher education institutions must reconnoiter how to combine humanistic qualities with virtual learning to guarantee student collaboration and engagement as well as face-to-face learning (Powell and McGuigan 2021).

Virtual learning also provides a competitive opportunity for higher education institutions to gain inclusiveness by catering their courses to a wide range of international students.

This also enables higher education institutions to pursue competitive brand positioning (Chatzoglou and Chatzoudes 2018). Virtual learning is utilized to create opportunities for transnational students. However, increasingly higher education institutions are using VL as a tool to engage disabled students who prefer limited movements (Gerrard 2007). There is a growing emphasis on developing a process model for building the space architecture, aiming at adequate student engagement.

### 2.2.3. Independent Learning

The modern education system recognises the importance of independent learning as part of its pedagogy. The COVID-19 pandemic further pushed higher education institutions to use independent learning as one of their main tools to overcome the challenges associated with not being able to utilise conventional learning environments. Utilising and encouraging independent learning practices has become necessary to cope with the rapidly changing current state of both scientific and information technology bodies of knowledge. Additionally, to stay abreast with the research and development needs.

The purpose of independent learning in a university environment is to enable students to (a) develop content-based knowledge, (b) gain technical know-how and (c) other capabilities (Sudirtha et al. 2021). In a postmodern society, one could argue that higher education institutions should design and develop content for independent learning and tailor the learning process according to the rapid changes both in the scientific and information technology disciplines- the industrialized world. Thus, we argue that developing an empirical model to establish a uniquely changing pattern becomes a fundamental necessity.

Independent learning among university students promotes entrepreneurial spirit (Tan 2013; Beeson 2016). Thus, there is a greater emphasis on this phenomenon (however, to establish independent learning among students effectively, higher education institutions require the right combination of tools and techniques, most importantly, its integration with the digital transformation strategy/blueprint (Carter et al. 2020). Notably, independent learning primarily happens through online channels in the current era; thus, it offers the luxury for the students to selectively utilize the tools and the sources (Lemmetty and Collin 2019). However, disseminating information and information exchange is the key to independent learning (Sudirtha et al. 2021).

### 2.2.4. Social Media-Saturated Education

The increasing use of social media-based education creates a new gap: how knowledge collaboration occurs in traditional society versus knowledge collaboration in social media saturated postmodern society (Abney et al. 2018). The integration of social media has become one of the key elements of the digital transformation of higher education institutions. It is viewed as a platform to engage students to develop a positive attitude about the globalised world (Carrigan and Jordan 2021). Faculties in higher education institutions also utilize social media for instruction and lecturing purposes using various technologies closely integrated with social media (García-Peñalvo 2021).

Specifically, there is a vibrant movement towards Facebook and LinkedIn (West et al. 2015). The gravitational movement is logical, given the incredible numbers of subscribers/users on Facebook; however, different results have been found for the usefulness, learning and development and engagement (Heiberger and Harper 2008; Kirschner and Karpinski 2010; Kolek and Saunders 2008). Higher education institutions have started to explore how Twitter might be used to develop engagement opportunities for students, faculty, and education communities (Kassens-Noor 2012; Rinaldo et al. 2011). For example, research indicates that Twitter is assessing a microblogging feature that facilitates educational dialogues delivered in just time/real-time (Junco et al. 2011).

The competitive advantages of social media for higher education institutions are:

a.    The competitiveness of the university is dependably exhibited by social media and there is a rising hype around it.

b.    Social media networks have produced unique values and benefits

c.   The content shared relatively affects the productivity of student groups
d.   Influence to build/enhance industrial knowledge/intelligence
e.   Collaborative learning is possible
f.   Consistent, people-to-people interaction led to convergence and divergence

Postmodernist scholars argue that the saturated use of saturated social media is a vibrant feature of postmodernism. The use of online connectivity and the effective use of social media have collectively enabled the modern generation to pursue knowledge uniquely. This phenomenon has allowed higher education institutions to explore new patterns via social media interactions to stimulate the learning process. Researchers have found that effective use of social media leads to critical thinking and student engagement (García-Peñalvo 2021).

### 2.3. Distributed Autonomous Organizations (DAO)

Higher education institutions have focused on digital transformation strategies to ensure futureproofing and maintenance of competition in global education. This highlights the importance of examining what is required to stay competitive and how does the competitive landscape change? We claim that DAO implementation is critical, and it must be held accountable for developing a digital transformation blueprint/regulating sustainable digital delivery using digital business models. The emergence of DAO leads to new educational opportunities, develop the digital economy, forms digital cooperation among higher education institutions and their distributed operations (Burkov 2020).

The rise of DAO also led to the rapid development of innovative technologies, digitalization of education, enhanced digitalization of society and increased the number of hardware devices connected through the Internet of Things. There is a potential that the connected hardware devices will lead to borderless higher education institutions. It will introduce a diverse range of needed digital education and learning skills and technologies. Forming such DAO is an educational business challenge. Additionally, DAO could be viewed as a direct substitute for the conventional educational delivery of traditional higher education institutions. Despite the criticality of DAO, there is also a need to regulate and optimize several processors due to increasing digital freedom and scaling security-related issues to deal with b both the short-term and long-term postmodern-educational challenges.

The introduction of Blockchain Technology has increased the scope, scale and practicality of DAO. The unique technology that underpins blockchain has broadened the efficient functioning of DAO, specifically in the education sector, where content (digital assets), selection of courses (individualism/consumerism) and the recognition (offering certificate of recognition) are performed over the distributed autonomous network. Thus, it becomes strategically important for higher education institutions to focus on DAO, which holds the accountability to build digital delivery advantage. DAO development in the education industry is achieved by monitoring and controlling using formalized rules. These rules are designed based on real-time performance indicators. Increasingly the process of managing and controlling the DAO is automated using digital transformation capabilities, software, digital technologies and rule engines are used (Kaal 2021; Virovets and Obushnyi 2020).

Scholars argue that DAOs will be the future of many global industries, thus, the education industry cannot be an exception. However, there are many arguments and discussions about the DAOs structure, delivery model and capital growth. It is believed that future educational opportunities will be created via the formation of DAOs, based on their digital transformational capabilities-digital interaction between higher education institutions/students. The efficiency of the functioning of the DAOs is believed to be based on their ability and flexibility to decentralise their governance. However, digitally distributed autonomous organisations' security, governance, regulation, and legislation are still underdeveloped areas (Yan et al. 2013).

*2.4. The Fundamental Model of Postmodern Education Delivery*

The derived fundamental model shown in Figure 7, highlights several propositions. The model also simplifies the approach and underlying notion of the research. It draws on the current literature and attempts to illustrate the methodology, analysis and conclusions through (a) forces of postmodern education parameters estimated (b) quantifying its impact on education and (c) the close association sustainable digital transformation enabling the delivery of education. Thus, it offers a unique perspective on the postmodern education phenomenon and sustainable digital transformation, which is strategically important for university education worldwide. Global education has proved to be noteworthy in the evolution of education and it is highly likely to be even more significant in the future amid globalization (Dlačić et al. 2013; Bagci and Celik 2018; Alalwan et al. 2021).

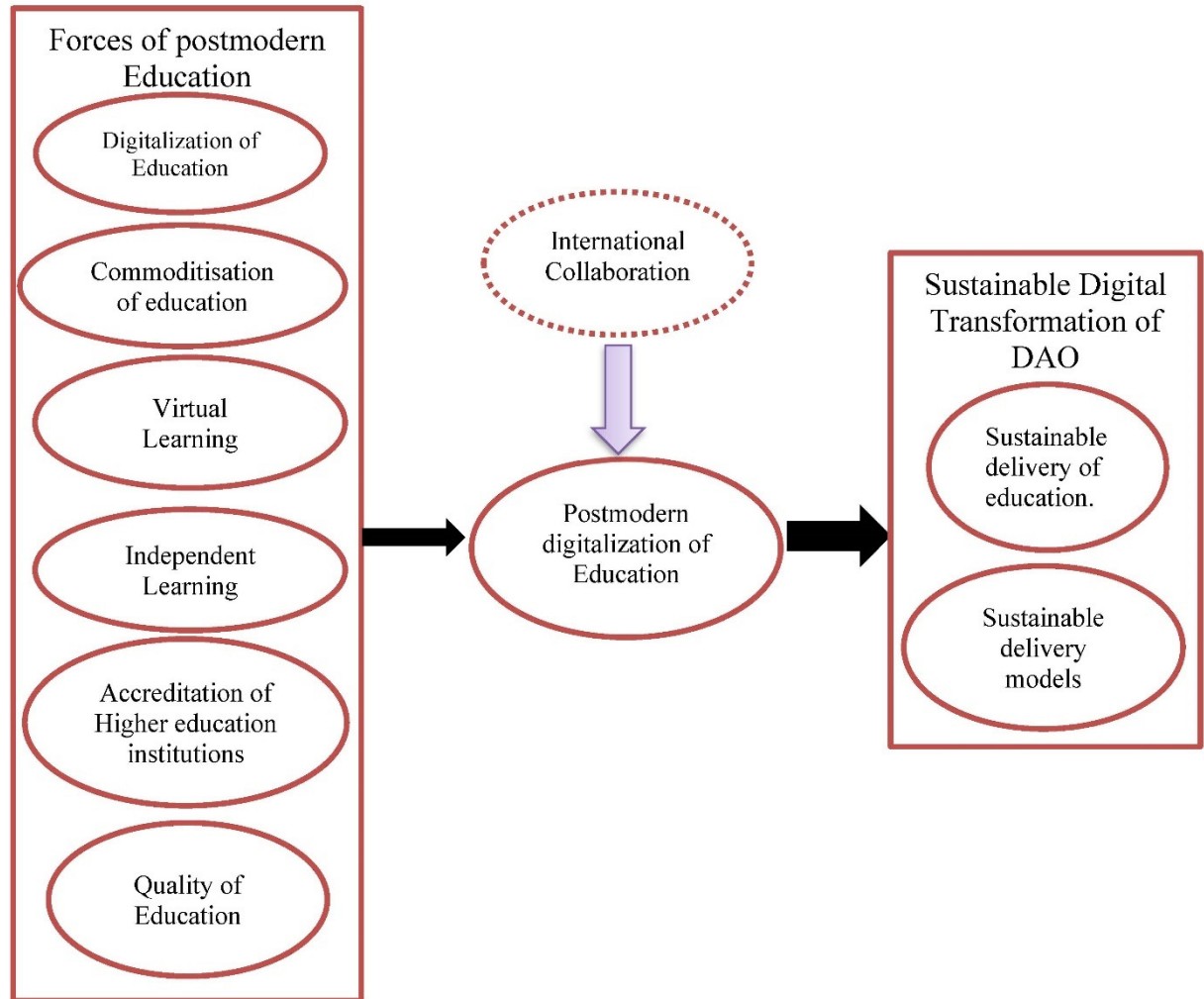

**Figure 7.** The nested fundamental model derived from the literature review. Source: Based on Dlačić et al. (2013); Bagci and Celik (2018) and Alalwan et al. (2021).

## 3. Methodology

Structural Equation Methodology (SEM) has been proposed to examine the impact of postmodernism and the university's ability to transform sustainable delivery of education using digital transformation capabilities. SEM offers a robust approach to examine the accumulated influence of postmodern educational forces (latent exogenous variables), the impact of postmodernism on education (proxy) and its impact on sustainable delivery capabilities (latent endogenous). Thus, the emphasis on understanding the accumulated impact using a quantitative approach, SEM using Confirmatory Factor Analysis, is recommended.

The SEM approach enables higher education institutions to examine the accumulated impact of postmodernism on the sustainable delivery of education via digital transformation. It is a robust statistical framework that is increasingly used across organizational research. It specifically enabled the researchers to test (a) latent variables, (b) measured variables and (c) the direct and indirect relationship in a structural model. The model identified (postmodern-digital) encapsulates both measurement and structural models. Thus, the need for the SEM method becomes inevitable. Figure 8 shows the authors' perspective regarding the examination of the impacts of postmodern digitalization. It is possible for future researchers to adopt either Confirmatory Factor analysis (CFA) or Exploratory Factor Analysis as the data analysis techniques on SEM. Thus, the interpretation of the data analysis is limited by the standard fit indices (model fit the data/data fit the model).

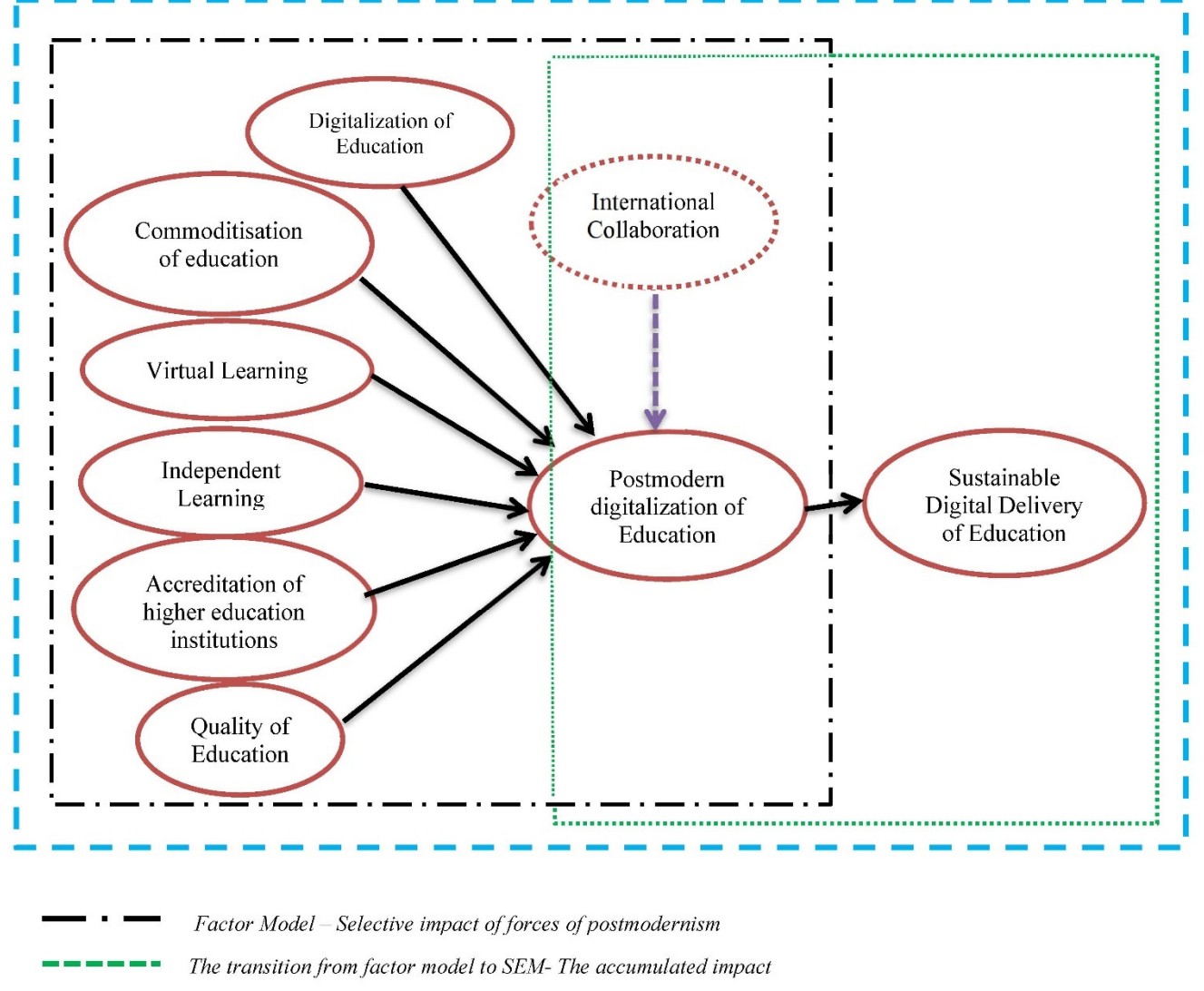

**Figure 8.** The measurements models and structural model of postmodern digital. Source: Based on Authors variable view.

This approach enables an organization to quantify (a) the individual impact of post-modern forces and (b) its collective impact on sustainable delivery; hence, transforming the same approach as a potential management information system becomes easy. Further, SEM also enables capturing the indirect effects among the variables (path models, mediation,

and moderation). This insight is critical to developing the narrative form of the knowledge of postmodernism.

The conceptual framework of the research (Figure 7) shows the projected association and relationship of postmodern forces of education. We view those as the antecedents of postmodernism of education. Then, it is linked to the influence of postmodernism on education, and as an outcome, how the sustainable delivery of education is impacted is examined by the conceptual model. Further, how the international collaboration of higher education institutions further amplifies the sustainable delivery of education is an important feature of the digital delivery of education. In the postmodern view, this could be viewed as a contextual parameter. Thus, the mediating mechanism is incorporated as one of the key elements of the conceptual framework.

## 4. Analysis

One of the primary tasks of higher education institutions is to provide informed knowledge to meet the demand in the wider community, engage in research that contributes to the body knowledge and economic factors such as making a profit. The process is significant, complex and evolving. Higher education institutions worldwide generally establish knowledge production and delivery by retaining recognized intellectuals and a dependable process. Digitalization, demand for industrialised knowledge, the dominance of information technology, commoditization, and the influence of postmodernism have enabled higher education institutions to re-engineer their education strategy to adapt to the rapidly changing environment by adopting innovative capabilities. Specifically, in the current era for a university to be competitive, we constructively argue that they require the integration of (a) an emergent strategy, (b) the integration of postmodern-digitalization and (c) developing a mechanism for sustainable delivery of education. We examine how integrating an emergent strategic approach with postmodern digitalization can improve the sustainable delivery of education. Scientific approaches can be formulated to test and validate this proposition. This is the centric argument/value addition of this paper. We believe that this approach would enable us to discover the systematic changes/differences triggered by postmodern digitalization in higher education institutions in a wider context (Holsberry 1981; Lyotard 1984; Arends 2014; Czainska 2009; Audebrand 2010).

Higher education institutions have adopted the emergent strategy approach to enable agile responsiveness and establish competitive advantages. It is creative, requires design thinking, dynamic and analytical in approach. The use of the emergent strategy is widely acknowledged. The characteristics of the emergent strategy enabled higher education institutions to gain distinct benefits, primarily; it enables higher education institutions to be flexible and agile in their approach to the delivery of education while fostering continuous learning and improvement. We argue that demonstrating agility for higher education institutions has become necessary to meet the need for the postmodern digitalization of education. Meaning the integration of postmodern digitalization needs an emergent approach to the portfolio of courses, particularly its design and delivery. As far as this research is concerned, it offers the luxury to the higher education institutions to deal with the business environment with agility (absorbing the influence of postmodernism) based on using competitive resources (digital transformation capabilities) to meet the purpose (sustainable delivery of education) (Fixson and Rao 2014; Du Toit and Verhoef 2018).

We propose using an emergent approach (Figure 9) for education strategy is innovative, incremental and complex to describe but a necessity. The intended strategy predicts the future based on the patterns of business activities from the past (Mintzberg 1987). Industrial knowledge is increasingly becoming a commodity. Particularly, information technology has gained accelerated growth as a service. The commoditized knowledge is forming new industries; this process has become a new trend among higher education institutions. By developing new knowledge, higher education institutions gain power and build competitive advantages. The mechanisation of knowledge is bound to affect higher education institutions when it becomes irrelevant or of no use (Lyotard 1984; Duvnjak et al. 2020).

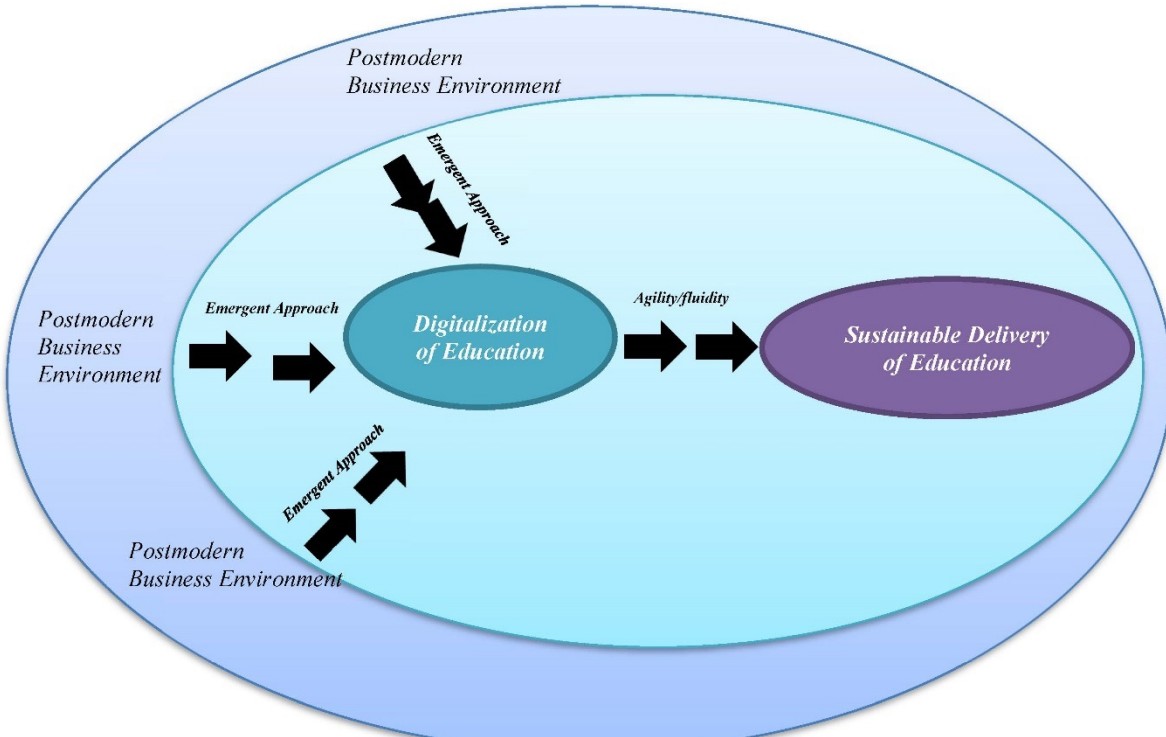

**Figure 9.** An emergent education strategy approach amid postmodernisms. Source: Based on Fixson and Rao (2014), Du Toit and Verhoef (2018) and Authors proposal (2022).

Higher education institutions' perspectives on the learning patterns and delivery of courses fall within the boundaries of the national education agenda, the national educational framework, and society's brain. Educational communication and transparency are directly related to the potential commercialisation of education. The higher education institutions' economic performance becomes the point of imperilling stability or standard. The gained capital and the monetary advantages are invested or pumped into new channels of multinationals. It could be observed that learning is going through, designed and retained in the same channels as money (Kane 2017).

Modern technology permits education to scale via a more agile approach, constantly producing scientific, technological and research driven knowledge, a typified feature of a postmodern society. It is noticeable that although higher education institutions show dissonance that scientific knowledge does not characterize the totality of knowledge production, it is also important to ensure a high emphasis on industrial research knowledge because of the demand conditions of global education. What will happen to the narrative form of knowledge production in higher education institutions continues to be a grey area. Despite our dissonance that the narrative form of knowledge cannot supersede the research-science-technology knowledge, the narrative form of knowledge is necessary to develop an implementable system body of knowledge (Lyotard 1984; Lacan 2019).

To cope effectively with the postmodern influence, higher education institutions have explored the case of education delivery. The power of computing to magnify the knowledge base of late digital transformation is used to amplify the sustainable delivery of education worldwide. The use of pedagogy-blended learning, independent learning, and virtual learning enhances the students' high use of digital education platforms. They have simply oversaturated education life. In a postmodern society, the bargaining power of the students relevant to selecting a course and pursuing it is high because of the interpretation and perception power of the customer. Additionally, many higher education institutions are more open about their commercial performance; the sustainability of the delivery is

determined by how many unities of a commodity (portfolio of courses) are sold to the students through digital channels.

What is a core (set of) proposition (s)? Sudden (late 20th or early 21st century) eruptions have fragmented the competitive landscape of higher education institutions. From a gradual, relatively smooth evolution landscape of them over the centuries, which Lyotard, for example, describes as modernistic, an era, described as postmodernity, arrived as a shock to businesses and governments. As it is often the case, the early heralds appeared in the arts: literature, criticism the graphic and performance arts (Lyotard 1984).

Interestingly, postmodernity was fostered, particularly within the humanities departments (i.e., working with sophisticated digital transformation technologies has become the new normal, business as usual). Academics fostered postmodern themes in the humanities both in action and reaction: that is, embracing postmodernism as an outlook and resisting it, and so creating a dialogue, on the other hand, promoters of postmodernism and the other opposes. Action and reaction create a dialogue that raises postmodernism to eminence with supporters and refusers. Early refusers were in the arts, and postmodernist trends became visible in the nano and digital technologies. The new technology erupted through the economics of scope and scale and network effects, which diffused the new technologies exponentially, simultaneously the variety of new products increased, their cost fell, and productivity in high-tech industries accelerated.

Again, ironically, higher education institutions were central to all these trends: central to its development and adoption. By adopting digital technologies operationally, they became networks. Strategically, though they remained hierarchies, governed from within by managerial hierarchies and governed from the outside by direct central governments and central government quangos and other regulatory bodies decided upon by higher education institutions themselves. Thus, we describe aspects of postmodernity: fragmentation and distribution system, operationally, flattened/consoles by the internal and external hierarchies, hierarchies that inhabited adaptive strategies.

Postmodernism relatively has provoked architectural layers and features in education. The influence of the postmodern is a phenomenological experience, representing the core thinking on central issues of postmodernity (Foss et al. 2021). Postmodernism rejects universality/single universal science base. It brings the advantage of producing multiple methodological approaches to the research process. In this background, if we want to discuss/approach the production/representation of knowledge, what methodologies will we select and apply? We believe SEM as a methodology offers a broad option to test the integrated influence of postmodernism in digital delivery. SEM integrates both exogenous and endogenous latent variables (forces of postmodern education and its outcome), universities' existence and how they can transform these rapid changes to their advantage and its observables using a statistical framework that is needed to test the accumulated impact of a postmodern variable on digital delivery. CFA on SEM enables the researchers to quantify with the high-loading observables, thus determining the factors that empirically influence postmodern-digital reality formation. Further, the use of CFA on SEM enables higher education institutions to smoothly transform the postmodern-digital reality into a management information system, using its core features (Bagci and Celik 2018; Alalwan et al. 2021). A theoretic approach postmodern-digital approach can improve the strategy formulation and implementation in education (Figure 10).

Pursuing sustainability in digital delivery amid modern, postmodern challenges is complex; it requires an incremental approach to delivery but identifying and determining the key elements of the delivery becomes critical to impart sustainable digital systems in higher education institutions. Thus, we suggest using the model of the Deming Cycle (Plan-Do-Check-Act) incrementally. Further, Glavič and Lukman (2007) developed an incremental model capturing the key elements aligned in Figure 11.

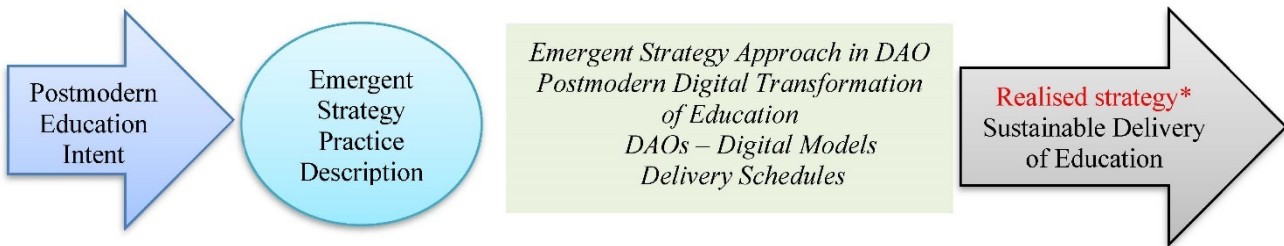

*\* only the emergent part of the realised strategy*

**Figure 10.** Emergent Strategy is Displayed as the Realised Strategy. Source: Based on (Foss et al. 2021).

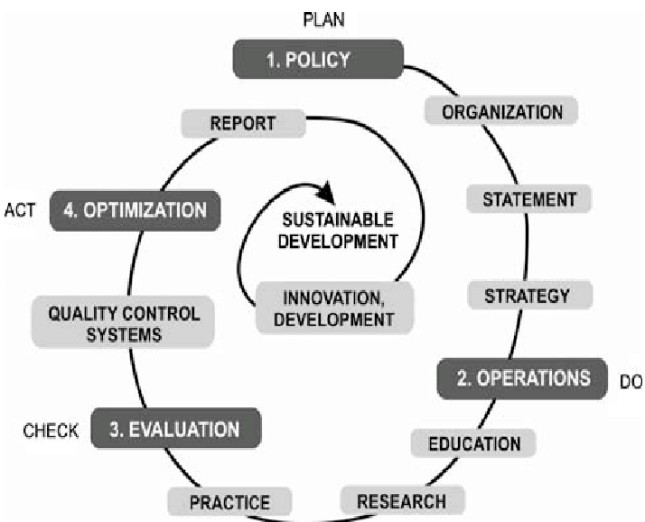

**Figure 11.** Process and Elements of a Sustainable University. Source: Based on Glavič and Lukman (2007). Source: www.google.com (accessed on 1 December 2022) based on Glavič and Lukman (2007).

We also view the incremental approach to gaining sustainable delivery as requiring time, tools and technique. However, defining an approach that discusses the variables and performance indicators is key to success. Based on the facts derived from the literature, we would like to suggest the approach shown in Figure 12 for sustainable postmodern-digital delivery.

The research findings aid the evolution of sustainable digital transformation practices in higher education by producing empirical insights into determining the impact of postmodernism and its association with sustainable education. It also highlights the strategic importance of using a sustainable digital transformation to generate and regulate sustainable education programmes (Demenko and Savina 2019). The paper also delivers fresh insight into the impactful postmodernism changes affecting higher education institutions' existences and how they can transform these rapid changes to their advantage. A different sense of education life creates a unique thinking style and attitude. Postmodernism and its influence have shaped education through the power of digital delivery. In turn, it has impacted sizeably the higher education life of global students. The main elements of postmodern culture and its influence, the importance of viewing through the lens of digital transformation, are integrated with the digital delivery of the educational programme in higher education institutions.

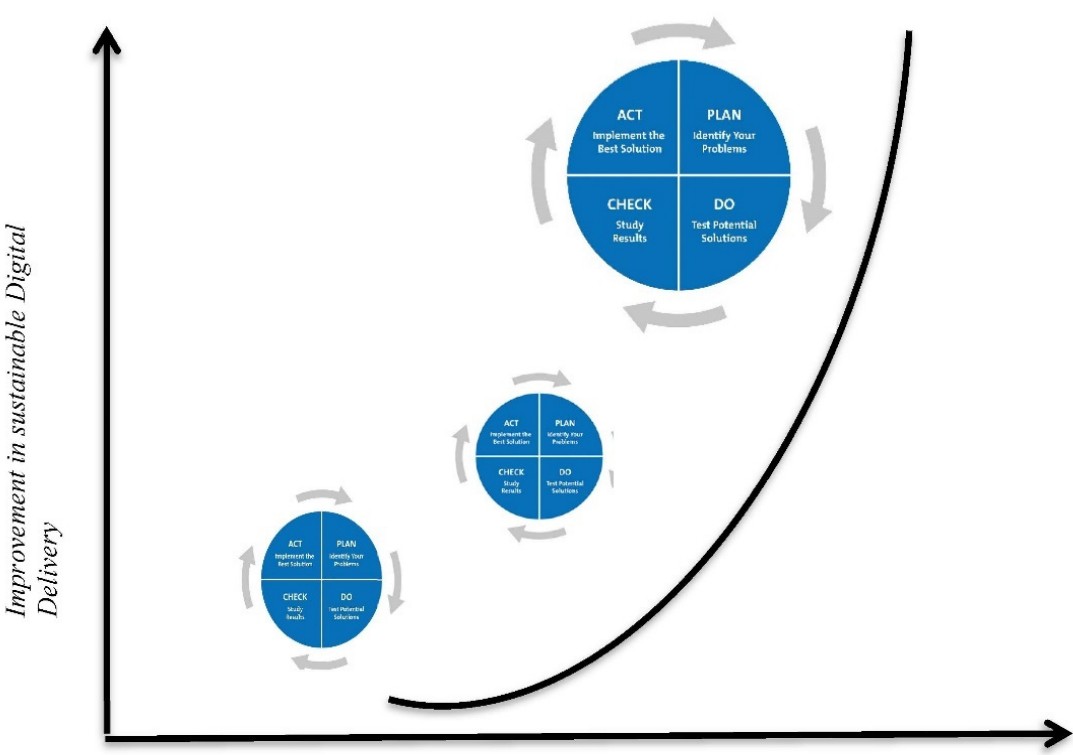

Technology independent over time

**Figure 12.** Incremental Improvement of Sustainable Digital Delivery. Source: Based on Demenko and Savina (2019).

## 5. Conclusions

This concept paper attempts to develop a theoretical model that embarks on how sustainable digital transformation as an educational force could be better utilized to achieve sustainability in higher education amid post-modernism. We propose a deductive research approach to examine this niche phenomenon using the SEM on CFA/EFA as the research strategy/data analysis technique. We claim that the traditional/analytical approach can produce a significant debacle because the approach is conventional, leaner, and premeditated. It also does not include the vibrant feature readily concerning the changing market condition. Emergent strategy produces a practical sequence of logic. We argue emergent approach to education strategy is essential to stay competitive in the rapidly changing globalized education sector.

The implementations of post-modernistic-emergent strategy models lead to new sustainable digital transformation capabilities in higher education and new education technologies, portfolios of courses, and policies. Universities have a role to play in the wider community and ultimately, they have a responsibility to lead the formation of a digital-oriented society. It implies that sustainable digital transformation in post-modernistic education is a necessity/unavoidable but not a luxury. This unique phenomenon put pressure on the applicability of the emergent strategy. It also means utilizing the sustainable digital transformation as a global platform in education to provide equal access to deliver quality education service is a significant challenge primarily because of the digital divide. Although this paper has discussed the significance of those phenomena to a greater extent, it is restricted by several limitations such as (a) it limits the applications of complex theoretical ideas (postmodernism/emergent strategy/digital transformation) to the education industry and (b) it focusses primarily on the technological, human and sustainable drivers in the higher education institutions.

The findings of this research answer our research questions on how postmodern digitalization impacts the sustainable delivery of education in emergent strategy. Our findings (literature/analysis) show that the intent for postmodern education is critical for formulating an emergent approach, explicating the integration of key activities of digital transformation is inevitable to enable the sustainable delivery of education. By cautiously integrating the connection between the influence of postmodernism on education, its digital enablement and sustainable delivery of education, our research makes a notable contribution by developing (a) a measurements model (b) a structural model and (c) offering a data analysis technique confirmatory factor analysis on SEM.

The implementation of emergent strategy requires postmodern education intent and autonomous strategic behaviour. Its integration with digital transformation to resolve the sustainable delivery of education is an approach to fill the global revenue gap as far as higher education institutions are concerned. Higher education institutions are under increasing pressure to regulate their research activities and scholarship to react to these changes.

Future researchers may focus on applying the suggested conceptual models in this paper to validate the practicality of generating beneficial outcomes for higher education institutions. Further, they also may examine how the implementation differs in two distinguished contexts to alter the impact of postmodernism on education delivery, specifically the moderating/mediating effect of contextual parameters. Demonstrating a positive attitude in adopting postmodernist changes in a sustainable digital transformation journey for higher education institutions is an absolute necessity to be future-equipped and stay competitive in education delivery. Such an integrated view of postmodernism on sustainable digital transformation offers a wide range of education programmes worldwide. However, there is a paucity and lack of clarity among educational institutions on how (a) postmodernism and (b) sustainable digital transformation impact, shape, and continuously improve educational delivery.

**Author Contributions:** All authors have contributed equally and substantially to the manuscript's concept and design. The team conceived the presented idea, developed the theory, verified the analytical methods, supervised the findings, discussed the results, and contributed to the final manuscript. All authors have read and agreed to the published version of the manuscript.

**Funding:** This research has no external funding.

**Institutional Review Board Statement:** Not applicable.

**Informed Consent Statement:** Not applicable.

**Data Availability Statement:** Not applicable.

**Conflicts of Interest:** The authors declare no conflict of interest.

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
