# Peer review of "Emergent Strategy in Higher Education: Postmodern Digital and the Future?"

_admsci, doi:10.3390/admsci12040196_

Round 1
Reviewer 1 Report
Dear Author(s),
Please find attached my comments and suggestions.
Kind regards,
The reviewer

Author Response
Please see the attachment below

Reviewer 2 Report
Dear Author(s),
Please find below my concerns and remarks regarding your manuscript proposal entitled "Emergent Strategy in Higher Education: Postmodern Digital and the Future?" sent to Administrative Sciences Journal.
My first remark refers to the type of the paper. In my version of the manuscript, on the first row (before the title of the paper), the type is "Concept Paper". Then, within the abstract, you say that "This research paper offers a solution and a scheme that amounts to a process for developing a...", meaning that your paper is a research paper.
I don't understand very well if your manuscript wants to be a concept paper or a research paper.
Please revise and clarify this important aspect regarding the type of the proposed paper.
The Introduction contains a lot of text, but it needs to be organized in a better manner because at this moment the readers don't understand some important aspects. Thus, I recommend you to clearly define and describe the following elements:
- the research gap;
- the research goal;
- the research question(s).
At the end of the Introduction section, please add distinct paragraphs for these aspects, so that the readers fully understand what you want to cover by your research proposal.
The figures must have titles without active verbs. For example, the title of figure 5 should be revised in something like "The emergence to postmodern-digital future". The same remark for the other titles of the figures. Please revise them.
The Literature Review section must be enriched with additional sources from the literature. I recommend you to include the following relevant and valuable references: https://doi.org/10.15388/infedu.2022.13, https://doi.org/10.1007/s10639-021-10739-1, https://doi.org/10.24818/ie2020.02.01, https://doi.org/10.3390/su13042023, https://doi.org/10.1088/1742-6596/1946/1/012021.
This way, you will define a better context for your research proposal.
The texts from figure 2 are not very clear. In my version of the manuscript, the rectangles are too small.
Please correct this editing aspect.
Under figure 3, please write the source of the image. If it is you own vision, please write "Source: Authors' proposal".
In section "3. Methodology" you say that you used SEM (Structural Equation Modelling). Normally, you should also present the synthetic results of the SEM: the path coefficients, the p-values, the influences of the constructs etc.
I recommend you to add these aspects in your manuscript, so that the readers see and understand the numerical results.
In the figure from page 14 you have a green line entitled "The transition from factor model to SEM - The accumulated impact".
But I don't see the origin and the destination of this line. It seems to be an editing issue because the figure appears to be incomplete.
Please revise and correct this aspect.
The font size from figure 12 must be enhanced. It is unreadable now.
Dear Author(s),
Please consider all the above remarks as being constructive recommendations in order to improve the general quality of your manuscript proposal.
Kind Regards!
Author Response
Please see the attachment below.

Round 2
Reviewer 2 Report
Dear Author(s),
I have read the revised version of the manuscript and I appreciate your effort to improve your manuscript proposal.
After reading and analyzing the new form of the article, I have only some minor additional recommendations which should be addressed:
- in the Conclusion section, please revise the sentence "...it has several limitations but not limited to...". It doesn't sound good.
- in the Literature Review section I recommend you to also include the following additional valuable references: https://doi.org/10.3390/s20113291, https://doi.org/10.1109/CogInfoCom50765.2020.9237840, https://doi.org/10.24818/ie2020.02.01, https://doi.org/10.1016/j.ijinfomgt.2020.102183. By including these sources from the literature, you will widen the general context of you research.
- please highlight some managerial implications regarding you research results. Here is the place where you can "sell" your results to the readers.
Kind Regards!
Author Response
Please see the attachment below.
